# Club Convergence of Economies’ Per Capita Carbon Emissions: Evidence from Countries That Proposed Carbon Neutrality

**DOI:** 10.3390/ijerph19148336

**Published:** 2022-07-08

**Authors:** Zhaofu Yang, Yongna Yuan, Yu Tan

**Affiliations:** School of Public Policy and Management, University of Chinese Academy of Sciences, Beijing 100049, China; yangzhaofu20@mails.ucas.ac.cn (Z.Y.); tanyu21@mails.ucas.ac.cn (Y.T.)

**Keywords:** carbon neutrality, per capita carbon emissions, club convergence, influencing factors, Kyoto Protocol

## Abstract

To achieve the long-term goals outlined in the Paris Agreement that address climate change, many countries have committed to carbon neutrality targets. The study of the characteristics and emissions trends of these economies is essential for the realistic formulation of accurate corresponding carbon neutral policies. In this study, we investigate the convergence characteristics of per capita carbon emissions (PCCEs) in 121 countries with carbon neutrality targets from 1990 to 2019 using a nonlinear time-varying factor model-based club convergence analysis, followed by an ordered logit model to explore the mechanism of convergence club formation. The results reveal three relevant findings. (1) Three convergence clubs for the PCCEs of countries with proposed carbon neutrality targets were evident, and the PCCEs of different convergence clubs converged in multiple steady-state levels along differing transition paths. (2) After the Kyoto Protocol came into effect, some developed countries were moved to the club with lower emissions levels, whereas some developing countries displayed elevated emissions, converging with the higher-level club. (3) It was shown that countries with higher initial emissions, energy intensity, industrial structure, and economic development levels are more likely to converge with higher-PCCEs clubs, whereas countries with higher urbanization levels are more likely to converge in clubs with lower PCCEs.

## 1. Introduction

Excessive CO_2_ emissions from human activities have a significant impact on energy, economy, and industrial security and also pose a considerable threat to human growth and the global ecosystem [1]. Subsequently, the achievement of carbon neutrality is an essential step in mitigating global warming, which has elicited an international consensus. The Paris Agreement proposed a long-term goal of combating climate change by maintaining a global average temperature rise within 2 °C of preindustrial levels by 2100 and aiming to keep this rise within 1.5 °C. To support this goal, it has been suggested that global carbon neutrality (net zero carbon emissions) is required [2]. The Intergovernmental Panel on Climate Change also called for the attainment of global carbon neutrality by approximately 2050 to maintain the target of keeping global warming within 1.5 °C by 2100. Heeding this call, many countries have committed to the goal of carbon neutrality through legal provisions and policy announcements. On 16 September 2020, the European Union (EU) announced that, by 2030, the EU’s greenhouse gas emissions will be reduced by at least 55% compared with 1990, and achieve carbon neutrality by 2050. In September 2020, China announced that it will reach peak carbon emissions by 2030, and carbon neutrality will be achieved by 2060 [3]. After China proposed its carbon peak and carbon neutrality goals, developed countries, including Japan, the United States (US), Canada, and South Korea, also announced political commitments to achieve carbon neutrality by 2050 [4]. According to the statistics of the Energy and Climate Think Tank (ECIU), as of May 2022, 128 countries, 116 regions, 234 cities, and 699 enterprises have established net zero emissions goals. Among them, Germany, Sweden, Portugal, Japan, France, Britain, South Korea, Canada, Spain, Ireland, Denmark, Hungary, New Zealand, and the EU have passed legislation to establish carbon neutrality targets.

Faced with such demanding commitments, no major economy has yet achieved carbon neutrality, and all of the countries that have committed to this goal are exploring path designs and strategies to achieve carbon neutrality [5,6]. Currently, both developed and developing countries that have declared carbon neutrality goals are working to achieve these [7]. There are considerable differences in the economic development levels, natural endowments, and industrial structures of these countries [8]. At the international level, there is no recognized applicable carbon neutral standard [9], and countries have vastly differing circumstances [10]. Due to earlier progress in industrialization in developed countries, most of these have already achieved their carbon peak; developing countries remain in the rapid economic development stage, while some African countries have social development levels far below the world average [11]. It has been argued that eliminating income inequality while reducing carbon emissions can boost the possibility of reaching carbon neutrality [12]. Consequently, for countries to achieve the agreed net zero emissions targets, a cluster analysis of various national emissions characteristics is imperative to uncover the real trends in each country, which will help countries set carbon neutral climate targets [13,14]. This knowledge is also beneficial in establishing carbon neutral alliances among countries with similar emissions trends for more effective international climate cooperation. Based on the above-mentioned concerns, in this study, we investigate whether the carbon emissions of countries that have pledged to achieve carbon neutrality converge to a common steady-state level or multiple steady-state levels. What factors influence the evolutionary behavior and transitional heterogeneity of carbon emissions? Understanding carbon emissions convergence patterns and the determinants of the convergence clusters identified is crucial for policymakers aiming to develop appropriate environmental policies [15].

The construct of convergence was first used to investigate changes in the income growth of countries or regions with different trajectories of initial economic development [16], introducing the concepts of *σ*-convergence, absolute *β*-convergence, conditional *β*-convergence, and club convergence [17]. *σ*-convergence refers to the tendency for differences in economic development levels between countries or regions to decrease over time [18]. Absolute *β*-convergence indicates that countries or regions with lower economic development levels have higher growth rates than those with higher economic development levels [19], and all eventually converge to a steady state. Conditional *β*-convergence assumes that the equilibrium level is correlated to various conditions associated with development, and in general, countries that are further away from the equilibrium level develop faster [20]. Club convergence refers to the idea that different regions, depending on their initial conditions (technology, preferences, and political systems), will form different “clubs” in development that converge in similar regions [21]. Club convergence has been found to explain the coexistence of developed regions with surrounding economically disadvantaged regions more accurately, which is prevalent in the process of regional economic growth [22].

Regarding the research related to national environmental concerns, in addition to studies concerning the convergence of economic growth, convergence analyses of carbon emissions, carbon intensity, and ecological footprints have rapidly developed [23,24]. Strazicich and List (2013) were the first to use panel unit root tests and cross-sectional regressions to analyze CO_2_ emissions convergence in 21 industrial countries [25]. Zhu et al. (2020) analyzed the patterns of club convergence in eastern, central, western, and northeastern China based on geographical location [26]. Nevertheless, this research method has certain limitations. *σ*-convergence, *β*-convergence, and club convergence are concepts intended for use in the investigation of long-term changes and fluctuations, and convergence may only be identified in steady-state empirical results [27]. Consequently, this is inconsistent with the reality that a country’s convergence process may be ongoing, in which case, the rejection of convergence does not reflect the convergence process itself [28]. Moreover, traditional club convergence is based on artificial divisions of geographic location or economic characteristics [29], groupings that often lack a scientific basis and ignore individual differences within groups [17]. These shortcomings render such studies’ findings much less credible.

In response to these issues, Phillips and Sul (2007, 2009) proposed club convergence analysis using a nonlinear time-varying factor model that considers each country’s transition path and growth to find convergence [30,31]. Phillips and Sul (2007) also developed a new club convergence clustering algorithm for the classification of the total sample into different convergence clubs based on emergent characteristics in the data [30]. Since the introduction of this club convergence method, it has been applied by numerous researchers. Among the research regarding environmental relevance, Panopoulou and Pantelidis (2009) first studied the convergence of CO_2_ emissions in 128 countries over the period during 1960–2003 and analyzed the characteristics of the two convergence clubs. Most studies have focused on the club convergence characteristics of EU member states, US states, and Chinese provinces [32]. Specifically, Morales-Lage et al. (2019) examined per capita carbon emissions (PCCEs) across different sectors in 28 EU countries, finding that PCCEs increased in the agricultural industry, while decreasing in industry and energy production, and different convergence patterns emerged in the energy subsector [28]. Apergis and Payne (2017) divided the 50 US states by sector (residential, commercial, industrial, transportation, and electricity) into corresponding clubs for the period during 1980–2013, highlighting the importance of tailoring emissions reduction policies to the state-specific emissions convergence pathways reflected in the club divisions [33]. Akram and Ali (2022) examine the club convergence of natural resource rents for 108 countries over the period during 1970–2019, revealing output inequality in natural resource rents across countries [17].

Some scholars have recently begun to investigate the factors that affect the formation of convergence clubs [34]. In particular, Parker and Bhatti (2020) used the logarithmic mean Divisia index decomposition method to analyze the influencing factors of three convergence clubs, demonstrating that per capita income is the most significant driver in the formation of convergence clubs [35]. Most of the existing research has used ordered logit models to analyze influencing factors [36,37]. Bhattacharya et al. (2020) used ordered logit models to demonstrate that both renewable energy consumption and increased urbanization increase the likelihood of belonging to a club with a low carbon emissions intensity [38].

The purpose of this study is to analyze the carbon emissions paths and steady-state levels of countries that have committed to carbon neutrality targets, to construct an international club inventory of differing carbon emissions patterns, and to further analyze the driving factors that affect the formation of convergence clubs. First, we combine the endogenous club identification method with a nonlinear time-varying factor model to analyze the convergence of PCCEs in countries that committed to carbon neutrality targets during 1990–2019, analyzing the evolutionary path of each convergence club using relative transition curves. Then, we examine the factors affecting the formation of convergence clubs using the ordered logit model. Finally, we propose corresponding policy recommendations based on our research results. The main contributions of this study are threefold. (1) The achievement of carbon neutrality is a major global issue; thus, this study is not limited to a cluster of developed countries or a region within a country or a union, but analyzes all of the countries that have committed to carbon neutrality targets. (2) We compare and analyze the changes in each club’s emissions transition paths in the periods before and after the Kyoto Protocol was brought into being. (3) Based on the insights revealed regarding the formation of convergence clubs, the influencing factors of club formation are also investigated.

The remainder of the paper is organized as follows: Section 2 focuses on the model theory and data sources, Section 3 presents the empirical analysis, and Section 4 draws conclusions and proposes corresponding policy recommendations.

## 2. Materials and Methods

### 2.1. Club Convergence Model

We used the club convergence method of Phillips and Sul (2007, 2009) to estimate the convergence of PCCEs [30,31], dividing countries into different groups, with countries within the same club converging on a common club trend. For the panel data used in this paper, *i* = 1, 2, …, *N*, *t* = 1, 2, …, *T*, where *N* and *T* represent the number of countries in the sample and the length of time, respectively.
(1)$$CE_{it}=g_{it}+a_{it}=\left(\frac{g_{it}+a_{it}}{\mu_{t}}\right)\mu_{t}=\delta_{it}\mu_{t}\;\;\;\;$$
where $CE_{it}$ denotes the PCCEs, $g_{it}\;$denotes the stable systematic common component in the data, and $a_{it}$ denotes the temporary heterogeneous component included in the data. The common trend is denoted by $\mu_{t}$. $\delta_{it}$ is a time-varying heterogeneity factor that captures the deviation of each country *i* from the common path defined by $\mu_{t}$. The following parameters are defined referencing Phillips and Sul (2007) [30]:(2)$$h_{it}=\frac{y_{it}}{N^{-1}\sum_{i}^{N}y_{it}}=\frac{\delta_{it}}{N^{-1}\sum_{i}^{N}\delta_{it}}$$
where $h_{it}\;$is a relative transition parameter to measure the relationship between the panel mean at time point *t* of the coefficient $\delta_{it}$, which reflects the different transition paths of individual *i* in achieving convergent equilibrium. This parameter has two characteristics: (1) the cross-sectional mean of $h_{it}$ is equal to 1, and (2) when *t* → ∞, the cross-sectional variance ($H_{t}$) converges to zero, namely:(3)$$H_{t}=\frac{1}{N}\sum_{i=1}^{N}{\left(h_{it}-1\right)}^{2}\to 0\;\;\;\;\;\;\;$$

According to Phillips and Sul (2007), the semiparametric form of $\delta_{it}$ is shown below [30]:(4)$$\delta_{it}=\delta_{i}+\frac{\sigma_{i}\Psi_{it}}{L\left(t\right)t^{\alpha }},\;t\geq 1,\;\sigma_{i}\geq 0,\;\mathrm{for}\;\mathrm{all}\;\mathrm{individual}\;$$
where $\Psi_{it}$ is an iid(0,1) weakly dependent random variable, and L(t) is a slowly varying function. *α* is a parameter indicating the rate of convergence. The null hypothesis of convergence under this specific form for $\delta_{it}$ is $H_{0}$: $\delta_{it}$ = δ (where *α* ≥ 0), as opposed to the alternative hypothesis, $H_{A}$: $\delta_{it}$ ≠ δ (for all *i* and *α* < 0). Phillips and Sul (2007) constructed the cross-sectional variance ratio $H_{1}$/$H_{t}$ and built the following regression model to test the convergence of the clubs [30]:(5)$$\log\left(\frac{H_{1}}{H_{t}}\right)-2\log\left(\log\left(t\right)\right)=\hat{c}+\hat{b}\log\left(t\right)+\hat{\mu }\;$$
where *t* = [*rT*], [*rT*] + 1, …, *T*, and *r* = 0.3 for small/medium sample sizes (*T* ≤ 50), with [*rT*] being the integer part of *rT* and $\hat{b}=2\hat{a}$ being the estimate of α in Equation (2). The *t*-statistic $t_{\hat{b}}$ of parameter $\hat{b}$ is calculated using the heteroskedasticity autocorrelation robust standard error (HAC). The original hypothesis is only rejected at the 5% significance level if $t_{\hat{b}}$ < −1.65; otherwise, the original hypothesis of the existence of convergence is accepted.

Subsequently, the null hypothesis that the entire panel converge does not preclude the existence of specific convergent subgroups was rejected. Phillips and Sul (2007) proposed a four-step clustering algorithm that allows researchers to endogenously identify converging clubs in a panel [30]:

Step 1 (cross-sectional ranking): the overall countries are ranked based on the descending order of the cross-sectional data of the most recent period.

Step 2 (core group selection): the *k* largest countries after sorting are selected to form a subgroup $G_{k},N>K>2$. The *t*-statistic of Equation (5) is then computed for $G_{k}$, and the core group size of $k^{*}$ is selected according to the following criterion:(6)$$k^{*}=arg\;max_{k}\left\{t_{k}\right\}\;\;\;\;subject\;to\;\;\;\;min\left\{t_{k}\right\}>1.65$$

Step 3 (screening members): A country outside of $G_{k}$ is chosen, it is added to $G_{k}$, and the regression of Equation (5) is performed again. If the *t*-statistic is greater than 0, then this country is considered a member of club $G_{k}$ and joins $G_{k}$; otherwise, it is excluded from $G_{k}$. The same procedure is repeated for each of the remaining individuals, finally deriving a club containing all of the convergent individuals.

Step 4 (recursion and stopping): a complementary group of countries that were not selected in the third step emerges. The regression of Equation (5) is performed. If they converge, a second convergent club is formed. If not, steps 1–3 are repeated to identify other converging clusters. If, in the end, some individuals are not included in any club, these individuals are considered to be divergent.

### 2.2. Ordered Logit Model

It was assumed that countries that committed to carbon neutrality goals could eventually form *i* convergence clubs. The countries in the *i* convergence clubs were assigned values from 1 to *n* according to the PCCEs of the convergence clubs and arranged from the highest to the lowest. The ordered logit model was then used to analyze the factors influencing the convergence of clubs among the countries that committed to carbon neutrality goals. The model is set up as follows:(7)$$y^{*}=\beta X+\varepsilon_{i},\;u|X\sim Logit\left(0,1\right)$$
(8)$$y=\{\begin{array}{c} \begin{array}{c} 1,\;\;\;if\;y^{*}\leq r_{0}\\ 2,\;\;\;if\;r_{0}<y^{*}\leq r_{1}\\ \vdots \end{array}\\ n,\;\;\;if\;y^{*}>r_{n-1} \end{array}$$
where *y** is an unobservable latent variable corresponding to *y* (the assigned value of the country-affiliated convergence club). $r_{0}<r_{1}<\cdots <r_{n-1}$ represents the critical value or threshold parameter obtained by estimation. The relationship between *y** and *y* depends on whether *y** is greater or smaller than the given critical value. *X* is a set of explanatory variables, *β* is their estimated coefficient, and *u* is the random error term. In this paper, we focused on the coefficient *β* to determine the effect of each explanatory variable on club convergence by the sign of the coefficient.

### 2.3. Data Sources

First, we identified all of the countries that committed to carbon neutrality targets based on the net zero emissions tracking table published by the ECIU and matched them using the Oxford University Our World in Data database. Due to a lack of data for some countries, we selected the PCCEs for 1990–2019 for the club convergence analysis for 121 countries that committed to carbon neutrality targets, accounting for approximately 84.5% of the total global carbon emissions.

An ordered logit model uses cross-sectional data; therefore, in this study, we selected the average of the corresponding explanatory variables from 1990 to 2019 from the World Bank database. The specific explanatory variables included the initial PCCEs (ln*pco*), expressed by the logarithm of each country’s per capita carbon emissions in 1990; industrial structure (*il*), expressed by the proportion of industrial added value in each country’s GDP; energy intensity (*ei*), expressed as the ratio of primary energy consumption to each country’s GDP; the degree of economic development (ln*pgdp*), expressed by the logarithm of each country’s per capita GDP; renewable energy development level (*re)*, expressed by the proportion of each country’s renewable energy in total energy consumption; urbanization level (*ur*), expressed by the proportion of each country’s urban population in the total population; forest cover rate (*fc*), expressed as the percentage of each country’s forest area to land area; and foreign direct investment (*fdi*), expressed as the percentage of each country’s foreign direct investment to GDP. Descriptive statistics for the full sample are shown in Table 1.

## 3. Results

### 3.1. Full Period Club Convergence Analysis

Table 2, Panel A presents the club convergence analysis of PCCEs for the countries that committed to carbon neutrality. The first line is a club convergence analysis for all 121 sample countries, and the *t*-statistic value of −12.127 was smaller than the critical value of −1.65, rejecting the hypothesis of an overall convergence in the change in carbon emissions per capita in the sample countries. Subsequently, we used the method of Phillips and Sul (2007) to determine whether potential clusters of clubs were evident in the sample. The results are shown in lines 2 to 5, indicating the existence of four potentially convergent clubs. The *t*-statistics for Initial Club 1 (15 countries), Initial Club 2 (17 countries), Initial Club 3 (22 countries), and Initial Club 4 (67 countries) were −0.689, 3.193, 1.390, and 1.580, respectively, all of which were greater than the critical value of −1.65, supporting the original hypothesis of the existence of convergent clubs.

Finally, as the method for identifying convergent clubs tends to overestimate the true number of clubs. We assessed whether neighboring clubs could merge into larger clubs. According to the results, the *t*-statistic values of −5.995 and −1.690 for the Initial Clubs 1 and 2 and Initial Clubs 3 and 4 were both smaller than −1.65, indicating that these two groups of clubs could not be merged, whereas the *t*-statistic value of 1.371 found in the Initial Clubs 2 and 3 merger test was greater than −1.65, indicating that these two clubs could acceptably be merged. Finally, we obtained three convergence clubs, which are presented in Table 2, Panel B.

To reflect the trend and the evolutionary path of each club’s PCCEs relative to the steady-state average more visually, we plotted the relative transition path curves for the three convergence clubs. According to Phillips and Sul (2007, 2009), if the transition parameter of a convergence club is greater than 1, the emissions level of that convergence club is higher than the steady-state average of the sample. The results are presented in Figure 1.

The three convergence clubs converged to different steady-state levels through different emissions pathways, with significant variations, as the emissions levels of Club 1 are well above the steady-state average, and Club 3’s emissions are well below the steady-state average. Specifically, there is a wide variation in the characteristics of the countries in Club 1, which mainly include countries with large total emissions, such as the United States; countries with high GDP per capita, such as Estonia and Iceland; and energy producing countries, such as Saudi Arabia, which are in Club 1 because their PCCEs are much larger than the steady-state average and do not exhibit a decreasing trend. Club 2 consists mainly of European and developing countries with slightly higher economic levels and transition parameters that are slightly greater than 1. Being in this club is due to the fact that the PCCEs of these countries are close to the steady-state average and this club shows a slow decreasing trend. Club 3 primarily includes developing countries with lower economic levels and those countries in this club have PCCEs well below the steady-state average. Club 3 displayed a rapid upward trend from 2008 and is closer to the steady-state average.

According to the results, to achieve the goal of carbon neutrality, countries should develop emissions reduction plans that are tailored to existing development levels and specific emissions convergence pathways. Countries in Club 1 have much larger emissions pathways than other countries, and these countries must reduce carbon emissions to transition to a steady-state average and achieve the climate goal of 1.5 °C or 2 °C. These countries also have special emissions and industrial characteristics, and to achieve carbon neutrality, countries must actively learn from the experiences of countries in the same club. The emissions levels of countries in Club 3 are very close to the steady state. Among them, European countries have a high level of economic development and are in an early stage of energy transition and low carbon development. Other countries should actively look to these countries for cooperation and collaborative exchange regarding challenges and solutions. The economic and industrial levels of countries in Club 3 have not yet reached the global average, and they are still in the rapid development stage. It is difficult for small countries to avoid the rapid increase in carbon emissions during the early stage of development, and they need to actively promote national economic development while also transitioning to carbon neutrality.

### 3.2. Phased Club Convergence Analysis

The time span of this study is relatively long, and the results of club convergence reflect the convergence process of PCCEs across countries over a 30-year period. The longer the time span of club convergence, the differences in PCCEs across countries are gradually ignored [30,31]. Since the beginning of the new century, countries, particularly developed Western countries, have made carbon emissions reduction an important policy goal. It is difficult to club convergence under longer time span to reflect the changes in carbon emission convergence paths in each country due to environmental policies. The Kyoto Protocol, adopted in December 1997, was the first international agreement in human history to pledge a limit to greenhouse gas emissions in the form of regulations, which came into force in 2005. The principle of the Kyoto Protocol is universal, but includes differentiated responsibilities, in which only developed countries set targets for emissions reduction. Although most countries failed to meet the targets set by the Kyoto Protocol within the stipulated time frame [39], the implementation has still had some effect on reducing carbon emissions and preventing worse emission levels from occurring [40,41]. More importantly, the Kyoto Protocol meant that the world paid more attention to climate change and contributed to the introduction of subsequent international environmental agreements, such as the Paris Agreement. Therefore, this article considers the entry into force of the Kyoto Protocol as a key event to observe the convergence changes of PCCEs. After the entry into force of the Kyoto Protocol, countries that signed the agreement may take different degrees of carbon reduction measures, which make the PCCEs convergence paths of these countries change, and countries whose relative emission levels are reduced to lower convergence clubs imply the adoption of effective carbon reduction policies, which can provide policy recommendations for other countries aiming to achieve carbon neutrality. Subsequently, in this study, we analyzed the club convergence of PCCEs in countries that committed to carbon neutral targets in the pre-Kyoto Protocol (PKP) period from 1990 to 2004 and the post-Kyoto Protocol (AKP) period from 2005 to 2019, presenting the results in Table 3.

Table 3, Panels A and B present the club convergence results of PCCEs in countries with carbon neutral targets in the PKP and AKP periods. According to the results of the club convergence analysis for all 121 sample countries in the PKP and AKP periods, the *t*-statistic values were less than the critical value of −1.65, rejecting the hypothesis of an overall convergence of PCCEs in the sample countries. Subsequently, eleven convergence clubs emerged in the PKP period, and seven convergence clubs were formed in the AKP period. Finally, following the club merger test, six convergence clubs were formed in the PKP period, and seven convergence clubs were formed in the AKP period.

The relative transition curves of the clubs in both periods are presented in Figure 2. Based on the emissions levels, in both periods, Club 1 is defined as Club SH, representing a cluster of clubs with carbon emissions that are much greater than the steady-state average. Club 2 is defined as Club H, representing a cluster of clubs with slightly larger transition paths than the steady-state average. Club 3 in the PKP period and Clubs 3 and 4 in the AKP period are defined as Club M, representing a cluster of clubs with emissions levels of approximately 1 that are gradually transitioning to the steady-state average. Club 4 in the PKP period and Club 5 in the AKP period are defined as Club L, representing a cluster of clubs with transition levels that are lower than the steady-state average. Clubs 5 and 6 in the PKP period and Clubs 6 and 7 in the AKP period are defined as SL clubs, representing a cluster of clubs with transition levels that are much lower than the steady-state average. Table 4 presents a comparison of the club convergence of PCCEs in each country in two periods to analyze the changes in emissions levels in each country after the Kyoto Protocol came into effect.

First, we analyzed countries that transitioned to clubs with higher emissions levels, which implied that the countries in this category switched to a transition path with higher relative emissions levels. Among them, Kazakhstan transitioned from Club H to Club SH, China and Malaysia transitioned from Club M to Club H, and Laos transitioned from Club SL to Club H. These developing countries are not among those required to reduce emissions under the Kyoto Protocol, and in these countries, high pollution and high energy consumption economic development patterns in the early stages of development have led to significant increases in emissions to levels much greater than the steady-state average. For these countries to achieve carbon neutrality, it is necessary to alter the existing economic development model to mitigate the momentum of rapid carbon emissions growth and actively learn and adopt relevant emissions reduction measures and policies from countries with decreasing carbon emissions levels. At the same time, a large number of developing countries transitioned from Club L to Club M. Bhutan transitioned from Club SL to Club M, and Cambodia from Club SL to Club L. The carbon emissions from the economic development of these countries have transitioned them from lower emissions trends to gradually approaching the steady-state average; however, because the initial emissions levels of these countries were low and did not reach much higher than the steady-state average after the Kyoto Protocol, to achieve carbon neutrality in these countries, a low-carbon economic development model must be adopted to advance the transition to the steady-state average of emissions while the economy is growing steadily. The transition to a steady-state average level of emissions will continue.

Notably, some countries have transitioned to clubs with lower emissions levels, indicating a potential transition path for this category of countries to achieve lower emissions levels. First, Luxembourg, Seychelles, and the US transitioned from Club SH to Club H. Because of the high initial emissions levels in these countries, emissions levels remain much larger than the steady-state average, although some emissions reduction effects are evident. Second, a considerable number of countries, most of which are developed European countries, have transitioned from Club H to Club M. This circumstance is assumed to have occurred because these countries adopted policies corresponding to the reduction in carbon emissions in accordance with the Kyoto Protocol, and because the economic crisis in Western countries generated a reduction in output, resulting in lower energy consumption and carbon emissions. Finally, it is possible that the economic development of some developing countries with a lower level of development has continued to decrease from Club L to Club SL due to domestic economic disruption.

Finally, Table 4 demonstrates that the AKP period displayed the largest number of members in Club M, which is composed of countries that are larger or smaller than the steady-state average but are gradually transitioning to the steady-state average, with a large number of developed and developing countries decreasing or increasing the transition to Club M. This suggests that the inequality of global carbon emissions distribution is gradually decreasing and most countries’ emissions are gradually transitioning to the steady-state average.

### 3.3. Analysis of the Factors Influencing the Formation of Convergence Clubs

In this section, we further investigated the drivers affecting the formation of convergence clubs using the ordered logit model; the results are presented in Table 5. Column 1 shows the results of the drivers for the full sample period, and columns 2 and 3 show the results of the drivers for PCCEs in the PKP and AKP periods, respectively.

Industrial structure is significantly negative at the 1% level, indicating that countries with a larger share of industry are more likely to cluster in the higher PCCEs club. Countries with higher levels of industrial development tend to consume more fossil energy and produce more carbon dioxide. Energy intensity is significantly negative at the 1% level, indicating that countries with higher energy intensities are more likely to cluster in the higher PCCEs club. This is mainly because higher energy intensity implies that more energy is consumed to produce each unit of GDP, which leads to more CO_2_ emissions. Notably, energy intensity is not significant during the PKP period, while it is significant during the AKP period. This may be because after the implementation of the Kyoto Protocol, developed countries prioritized the reduction in carbon emissions by reducing energy intensity, causing the energy intensity impact club to converge much more. The level of economic development is significantly negative at the 10% level, indicating that countries with higher levels of economic development are more likely to cluster in clubs with higher PCCEs. At the same time, the level of economic development is not significant in the AKP period, which may be due to the fact that large carbon emissions tend to accompany rapid economic development in countries. Urbanization is significantly positive at the 10% level, indicating that countries with higher levels of urbanization are more likely to cluster in the lower PCCEs club. Governments and citizens in countries with higher urbanization levels tend to have more stringent environmental quality requirements, making it possible to independently take steps to reduce carbon emissions to maintain a better environment in these countries. The initial carbon emissions level is significantly negative at the 1% level, indicating that countries with higher initial carbon emissions are more likely to cluster in the higher PCCEs club. The insignificant coefficient of renewable energy development may be due to the fact that renewable energy is in its infancy and cannot affect the convergence trend of PCCEs. Additionally, the coefficients of forest cover and foreign direct investment are insignificant, indicating that these variables do not affect the club convergence of PCCEs.

From the above empirical results, we can see that reducing industrial structure and energy intensity is an effective measure to reduce national PCCEs and optimize the convergence path of carbon emissions. Additionally, with the entry into force of the Kyoto Protocol, the degree of economic development level affecting PCCEs gradually weakens, and energy intensity gradually becomes an important factor affecting the convergence path of PCCEs.

## 4. Conclusions and Policy Recommendations

Based on the club convergence method with nonlinear time-varying factors, we investigated the convergence characteristics of the PCCEs of 121 countries that committed to carbon neutrality targets globally, conducting a comparative study on the changes in emissions transition paths in each country before and after the Kyoto Protocol came into effect in 2005. On this basis, an ordered logit model was applied to analyze the factors affecting the formation of convergence clubs.

The results of this study yield three notable findings. (1) Three convergence clubs were evident regarding the PCCEs of the countries that committed to carbon neutrality. Different clubs converged to multiple steady-state levels along different transition paths, implying that countries in various clubs exhibited different convergence behaviors. Club 1 primarily included high emitters, developed countries, and energy-intensive countries with emissions levels that are well above the steady-state average and remain stable. Club 2 predominantly included developed and developing countries with high economic levels and emissions levels that are slightly above the stable average and remain stable. Club 3 primarily included developing countries with lower economic levels and emissions levels that are far below the stable average and exhibit a rapid upward trend. (2) The clubs in the whole sample were almost perfectly divided into the periods before and after the Kyoto Protocol came into being, with six convergence clubs in the PKP period and seven convergence clubs in the AKP period. As developing countries, such as Kazakhstan, China, and Malaysia, are not subject to the emissions reductions required by the Kyoto Protocol, these countries have switched to a transition path with higher emissions levels. Some developed countries have adopted corresponding emissions reduction measures, switching to a transition path with lower emissions levels. At the same time, most countries’ carbon emissions are gradually transitioning to the steady-state average level, and the inequality in the distribution of global carbon emissions is gradually dissipating. (3) The results of the analysis of factors affecting the convergence of clubs indicate that countries with higher initial emissions, energy intensity, industrial structure, and economic development levels are more likely to converge to clubs with higher PCCEs, and countries with higher urbanization levels are more likely to converge to clubs with lower PCCEs. Renewable energy development, forest cover and foreign direct investment do not affect the results of PCCEs club convergence.

Based on the above-mentioned findings, we propose three policy recommendations related to achieving carbon neutrality. First, each country should develop strategic carbon neutrality policies according to the emissions path of the club to which it belongs. Countries in clubs with higher emissions should reduce carbon emissions as soon as possible by developing and implementing low-carbon technologies, improving energy efficiency, reducing their consumption of traditional fossil energy, and reducing energy intensity. Countries in clubs with lower emissions are less developed and less industrialized, and the increase in emissions observed is mostly due to rapid economic development; thus, they need to maintain sustainable economic growth while achieving carbon neutrality. Each country is encouraged to actively learn from the carbon neutral experiences of countries in the same club and take stronger measures to move toward the goal of carbon neutrality while stabilizing the economy. Second, developed and developing countries have common, but differentiated, responsibilities. The historical cumulative carbon emissions of developed countries are much larger than those of developing countries [42], and most developing countries that have proposed carbon neutrality targets have unavoidable carbon emissions due to economic development. Therefore, a global carbon offset mechanism can be established, and developed countries that have achieved sustainable economies should enhance climate finance [43,44] and provide more technologies to help developing countries reduce carbon emissions. Third, the development of solutions for carbon neutrality requires global action; therefore, members of various clubs can promote global carbon neutrality by strengthening exchanges, cooperation, and collaboration in low-carbon technologies, new energy, and related fields, establishing regional carbon emissions trading markets, and setting up climate alliances.

Although this study offers some degree of improvement and innovation in research methods and research content, some limitations remain. For example, among the countries that proposed carbon neutrality targets, some countries’ emissions data are unavailable, and therefore, these countries had to be excluded from our empirical evaluation. On the other hand, as this paper includes many developing countries with insufficient data, it is difficult to further adopt a more detailed and diverse perspective to analyze the factors affecting club convergence. In addition, with the abundance of data and the extension of the period in future studies, the impact of international agreements such as the Paris Agreement on carbon emissions in each country after their implementation can be fully studied.

## Figures and Tables

**Figure 1 ijerph-19-08336-f001:**
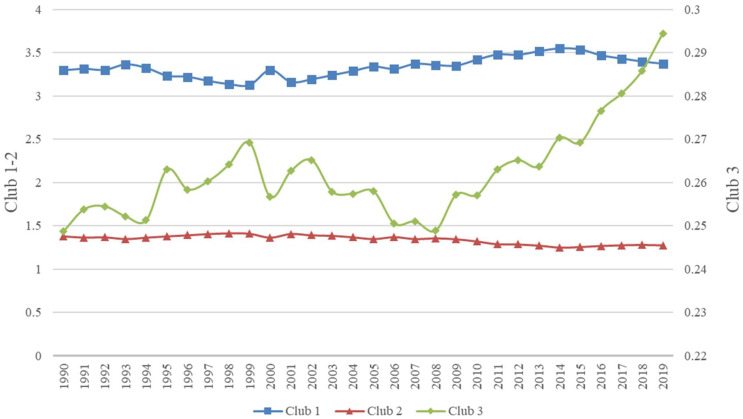
Relative transition path curve of convergence clubs.

**Figure 2 ijerph-19-08336-f002:**
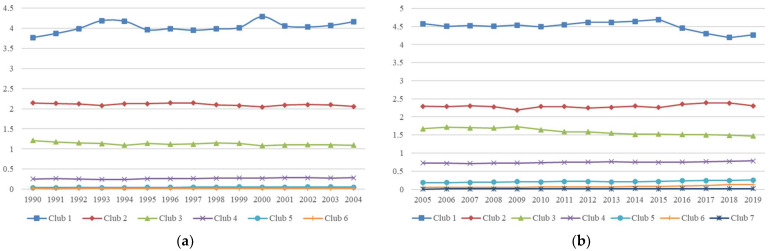
(**a**) Relative transition path curves of convergence clubs in AKP period; (**b**) relative transition path curves of convergence clubs in PKP period.

**Table 1 ijerph-19-08336-t001:** Descriptive statistics of variables.

VarName	Mean	Std. Dev	Definition
*pco*	4.753	5.814	Per capita carbon emissions
*il*	24.376	9.148	Proportion of industrial value added to GDP
*ei*	6.726	4.796	Proportion of primary energy consumption to GDP
ln*pgdp*	8.429	1.582	Logarithm of GDP per capita
*re*	36.520	31.585	Proportion of renewable energy to total energy consumption
*fc*	34.093	23.592	Proportion of forest area to land area
*ur*	53.339	24.512	Proportion of urban population to total population
*fdi*	4.861	7.953	Proportion of foreign direct investment to GDP
ln*pco*	0.338	1.893	Logarithm of initial per capita carbon emissions

**Table 2 ijerph-19-08336-t002:** Club convergence results.

**Panel A**					
**Club**	$\hat{b}$ **-Coefficient**	***t*-Statistic**	**Club Merging**	$\hat{b}$ **-Coefficient**	***t*-Statistic**
Full sample	−0.601 *	−12.127			
Club 1 (15)	−0.061	−0.689	Club1 + Club2	−0.435 *	−5.995
Club 2 (17)	0.403	3.193	Club2 + Club3	0.218	1.371
Club 3 (22)	0.382	1.390	Club3 + Club4	−0.169 *	−1.690
Club 4 (67)	0.134	1.580			
**Panel B**					
**Club**	$\hat{b}$ **-Coefficient**	***t*-Statistic**	**Countries**		
Club 1 (15)	−0.061	−0.689	Australia, Bahrain, Canada, China, Estonia, Iceland, Kazakhstan, Luxembourg, Malaysia, Russia, Saudi Arabia, South Korea, Trinidad and Tobago, United Arab Emirates, United States		
Club 2 (39)	0.218	1.371	Antigua and Barbuda, Argentina, Austria, Bahamas, Barbados, Belgium, Bulgaria, Chile, Croatia, Cyprus, Denmark, Finland, Germany, Greece, Hungary, Ireland, Israel, Italy, Japan, Laos, Latvia, Lebanon, Lithuania, Maldives, Mauritius, New Zealand, Panama, Seychelles, Singapore, Slovakia, Slovenia, South Africa, Spain, Suriname, Thailand, Turkey, Ukraine, United Kingdom, Vietnam		
Club 3 (67)	0.134	1.580	Angola, Armenia, Bangladesh, Belize, Benin, Bhutan, Brazil, Burkina Faso, Burundi, Cambodia, Cape Verde, Central African Republic, Chad, Colombia, Comoros, Costa Rica, Dominican Republic, Ecuador, Ethiopia, Fiji, France, Gambia, Grenada, Guinea, Guinea-Bissau, Guyana, Haiti, India, Indonesia, Jamaica, Kiribati, Liberia, Madagascar, Malawi, Mali, Malta, Mauritania, Mozambique, Myanmar, Nauru, Nepal, Nicaragua, Niger, Nigeria, Pakistan, Papua New Guinea, Peru, Portugal, Rwanda, Saint Vincent and the Grenadines, Samoa, Sao Tome and Principe, Senegal, Sierra Leone, South Sudan, Sri Lanka, Sudan, Sweden, Switzerland, Tanzania, Togo, Tonga, Uganda, Uruguay, Vanuatu, Yemen, Zambia		

Note: 1. * represents the rejection of the null hypothesis of convergence at the 5% level. 2. Numbers in parentheses represent the number of countries in a club.

**Table 3 ijerph-19-08336-t003:** Club convergence results before and after the Kyoto Protocol came into effect.

**Panel A: 1990–2004**								
**Initial Clubs**			**Club Merging Test**			**Final Clubs**		
	$\hat{b}$ **-Coefficient**	***t*-Statistic**		$\hat{b}$ **-Coefficient**	***t*-Statistic**		$\hat{b}$ **-Coefficient**	***t*-Statistic**
Full sample	−0.806 *	−25.716						
Club 1 (9)	−0.057	−0.374	Club 1 + 2	−0.613 *	−6.969	Club 1 (9)	−0.057	−0.374
Club 2 (22)	0.096	0.728	Club 2 + 3	−0.570 *	−12.208	Club 2 (22)	0.096	0.728
Club 3 (23)	−0.081	−1.202	Club 3 + 4	−0.363 *	−19.831	Club 3 (23)	−0.081	−1.202
Club 4 (12)	0.692	3.073	Club 4 + 5	−0.024	−0.162	Club 4 (35)	−0.194	−1.401
Club 5 (16)	0.367	3.406	Club 5 + 6	0.258	2.397	Club 5 (14)	−0.401	−0.959
Club 6 (3)	0.657	2.750	Club 6 + 7	−0.620 *	−3.311	Club 6 (18)	−0.082	−0.488
Club 7 (4)	−0.005	−0.009	Club 7 + 8	−0.186	−0.339			
Club 8 (2)	0.071	0.082	Club 8 + 9	−0.050	−0.104			
Club 9 (8)	0.703	1.232	Club 9 + 10	0.184	0.495			
Club 10 (4)	1.655	4.465	Club 10 + 11	−0.511 *	−8.813			
Club 11 (18)	−0.082	−0.488						
**Panel B: 2005–2019**								
**Initial Clubs**			**Club Merging Test**			**Final Clubs**		
	$\hat{b}$ **-Coefficient**	***t*-Statistic**		$\hat{b}$ **-Coefficient**	***t*-Statistic**		$\hat{b}$ **-Coefficient**	***t*-Statistic**
Full sample	−0.782 *	−14.216						
Club 1 (7)	−0.076	−0.332	Club1 + 2	−0.457 *	−3.554	Club 1 (7)	−0.076	−0.332
Club 2 (12)	−0.062	−0.411	Club2 + 3	−0.367 *	−3.618	Club 2 (12)	−0.062	−0.411
Club 3 (17)	0.306	2.652	Club3 + 4	−0.229 *	−4.646	Club 3 (17)	0.306	2.652
Club 4 (38)	−0.006	−0.055	Club4 + 5	−0.265 *	−2.577	Club 4 (38)	−0.006	−0.055
Club 5 (13)	0.288	1.461	Club5 + 6	−0.444 *	−6.653	Club 5 (13)	0.288	1.461
Club 6 (29)	−0.119	−1.348	Club6 + 7	−0.283 *	−3.948	Club 6 (29)	−0.119	−1.348
Club 7 (5)	0.029	0.141				Club 7 (5)	0.029	0.141

Note: 1. * represents the rejection of the null hypothesis of convergence at the 5% level. 2. Numbers in parentheses represent the number of countries in a club.

**Table 4 ijerph-19-08336-t004:** Comparison of convergence club groups.

		2005–2019				
		SH Club	H Club	M Club	L Club	SL Club
1990–2004	SH Club	Australia, Bahrain, Canada, Saudi Arabia, Trinidad and Tobago, United Arab Emirates	Luxembourg, Seychelles, United States			
	H Club	Kazakhstan	Estonia, Germany, Iceland, Japan, Russia, South Korea	Austria, Belgium, Cyprus, Finland, Greece, Ireland, Israel, New Zealand, Singapore Slovenia, South Africa, Denmark, Italy, Spain, United Kingdom		
	M Club		China, Malaysia	Antigua and Barbuda, Bulgaria, Lithuania, Nauru, Slovakia, Turkey, Argentina, Bahamas, Barbados, Chile, Croatia, France, Hungary, Jamaica, Lebanon, Malta, Portugal, Sweden, Switzerland, Thailand, Ukraine		
	L Club			Armenia Brazil Colombia Dominican Republic Eritrea Fiji Grenada Guyana India Indonesia Latvia Maldives Mauritius Panama Saint Vincent and the Grenadines Suriname Tonga Vietnam	Angola, Belize, Cape Verde, Costa Rica, Mauritania, Nicaragua, Pakistan, Papua New Guinea, Peru, Samoa, Sri Lanka, Uruguay	Kiribati, Nigeria, Sao Tome and Principe, Senegal, Yemen
	SL Club		Laos	Bhutan	Cambodia	Bangladesh, Benin, Comoros, Gambia, Guinea, Ethiopia, Haiti, Myanmar. Sudan, Togo, Vanuatu, Zambia, Burkina Faso, Guinea-Bissau, Liberia, Madagascar, Mali, Mozambique, Nepal, Niger, Sierra, Leone, South Sudan, Tanzania, Uganda, Burundi, Central African Republic, Chad, Malawi, Rwanda

**Table 5 ijerph-19-08336-t005:** The estimated results of the ordered logit model.

	(1)	(2)	(3)
Variables	1990–2019	1990–2004	2005–2019
*il*	−0.0836 ***	−0.111 ***	−0.0950 ***
	(0.0276)	(0.0329)	(0.0317)
*ei*	−0.241 ***	0.0881	−0.187 *
	(0.0930)	(0.0742)	(0.107)
ln*pgdp*	−0.767 *	−0.916 **	0.163
	(0.449)	(0.415)	(0.322)
*re*	0.0263	0.00644	0.0254
	(0.0205)	(0.0240)	(0.0171)
*fc*	0.0103	0.0161	−0.00851
	(0.0115)	(0.0145)	(0.00908)
*ur*	0.0269 *	0.0343 *	0.0210 *
	(0.0140)	(0.0181)	(0.0113)
*fdi*	0.0270	0.0560	−0.00851
	(0.0239)	(0.0774)	(0.00882)
ln*pco*	−1.066 ***	−4.253 ***	−3.751 ***
	(0.357)	(1.107)	(0.808)
Observations	121	121	121
Log likelihood	−57.705341	−60.999877	−87.799987
Pseudo R^2^	0.5047	0.7025	0.5819

Notes: robust standard errors in parentheses; *** *p* < 0.01, ** *p* < 0.05, * *p* < 0.1.

## Data Availability

The data presented in this study are available on request from the author. The data are not publicly available due to privacy. Images employed for the study will be available online for readers.

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
