# Peer review of "Club Convergence of Economies’ Per Capita Carbon Emissions: Evidence from Countries That Proposed Carbon Neutrality"

_ijerph, 2022, doi:10.3390/ijerph19148336_

Round 1

Reviewer 1 Report

A well written paper with original investigation. I would recommend minor editing of English before publication. 

Author Response

Response to Reviewer 1 Comments

Point 1: A well written paper with original investigation. I would recommend minor editing of English before publication..

Response : Thank you for your suggestion. This article has been edited in English provided by MDPI

Reviewer 2 Report

It is clear that decarbonisation strategies will differ from country to country. There is no indication in the text (especially in policy recommendations) that carbon emissions should be calculated taking into account past emissions. Countries that have achieved a high level of economic development should therefore invest (pay) for reducing emissions in countries that are just entering development paths. Perhaps a global compensation system would be needed.

Author Response

Response to Reviewer 2 Comments

Point 1: It is clear that decarbonisation strategies will differ from country to country. There is no indication in the text (especially in policy recommendations) that carbon emissions should be calculated taking into account past emissions. Countries that have achieved a high level of economic development should therefore invest (pay) for reducing emissions in countries that are just entering development paths. Perhaps a global compensation system would be needed.

Response : Thank you for the suggestion. We have made relevant policy recommendations based on the historical carbon emissions you mentioned and the findings of this paper. Please see P491--497.

Reviewer 3 Report

Review report of the manuscript IJERPH 1756364, with the title ‘Club convergence of economies' per capita carbon emissions: Evidence from countries that proposed carbon neutrality’.

Manuscript IJERPH 1756364 presents a mathematical model based on the economic theory of convergence applied to per capita carbon emissions (PCCE) of 121 countries. Both the convergence model and its application to carbon emissions are not novel issues. Based on the paper objectives (lines 67 – 72) which include the goal of “Understanding carbon emissions convergence patterns and the determinants of identified convergence clusters”, this manuscript suitability to the special issue "Decision Models for Sustainable Development in the Carbon Neutrality Era" appears to be fair. Nevertheless, Sustainable Development is not an issue addressed in this manuscript. Besides GDP and PCCE, other possible variables used for the model application in this study are not disclosed in this manuscript and no justification for this presented.

English writing, especially in the introduction section need improvement to become coherent and scientifically correct, using the adequate references. Along the paper there is the need for accuracy, such as the following examples:

the European Union and Europe are not the same entities and represent quite different geographic areas and countries (lines 37 to 40).

Panel and Plan (Tables 1 and 2 and lines 244, 262, 315) are not interchangeable as they have different meanings.

Concerning the references nearly one third of the references are about China which is rather curious considering that the study involves 121 countries. What was the reason for not including Norway, perhaps the most highly developed country in the world?

Paris Agreement is mentioned in the “abstract” and after in the introduction but, then (lines 144 to 146) authors write: “(2) We compare and analyze the changes in each club’s emissions transition paths in the periods before and after the Kyoto Protocol came into force.” Although several of the Kyoto Protocol mechanisms are still in force, such as the carbon market, for example, the Kyoto Protocol failed to cut GHG emissions and perished with the Doha Amendment and all the other attempts until reaching the Paris Agreement. What are the reasons for not considering the Paris Agreement national commitments (NDC) and instead of insisting to explore the model with the old Kyoto Protocol? In lines 309 to 311 authors write “We posit that the convergence club division of PCCE in each country may change significantly in 2005, and the countries with reduced relative emissions levels can offer an advantageous policy reference for other countries.” What is the scientific base for such assumption knowing the history of Kyoto Protocol and its outcomes?

On the other hand, in lines 301 to 304 authors state “Based on the club convergence of PCCE for each country in Figure 1, it is difficult to reflect the change in the carbon emissions levels among countries because of the long time span.” Please explain the difficulty.

Concerning results: they are presented in a quite confusing manner expanding from an initial group of 4 ou 3 clusters presented in Table 1, passing through 11 convergence clubs (Table 2 – panel A) to a final set of 7 clubs. In all clubs, countries with quite different characteristics are positioned in the same group and this is not discussed in the paper. For example, Luxembourg is geographically quite a small country but a very high standard of living. Luxembourg is club 1 (table 1) alongside with Ethiopia, Iceland or Trinidad and Tobago. What relates these so different countries? In lines 273 to 275 authors write “Specifically, Club 1 primarily includes countries with large total carbon emissions, such as the US, high per capita GDP countries, such as the Netherlands, and energy producing countries, such as Saudi Arabia.” However, was the Netherlands part of the study? Club 1 of Table 1 does not include the Netherlands.

The analysis of the factors that were found to influence the clusters do not present any novelty: industrialization, energy intensity and GDP are factors that are well-known to positively influence carbon emissions.

Author Response

Response to Reviewer 3 Comments

Point 1: Besides GDP and PCCE, other possible variables used for the model application in this study are not disclosed in this manuscript and no justification for this presented.

Response : Thank you for your suggestion. The carbon emissions per capita used for modeling in this study and other variables are disclosed in Table 1. Descriptive statistics of variables.

Point 2: English writing, especially in the introduction section need improvement to become coherent and scientifically correct, using the adequate references. Along the paper there is the need for accuracy, such as the following examples:the European Union and Europe are not the same entities and represent quite different geographic areas and countries (lines 37 to 40).Panel and Plan (Tables 1 and 2 and lines 244, 262, 315) are not interchangeable as they have different meanings.

Response : Thank you for your suggestion. We have thoroughly checked this article to eliminate formatting errors. Meanwhile, this article has been edited in English provided by MDPI.

Point 3: Concerning the references nearly one third of the references are about China which is rather curious considering that the study involves 121 countries. What was the reason for not including Norway, perhaps the most highly developed country in the world?

Response : Thank you for your suggestion. We have revised some of the research literature in this paper to conform to the research requirements of this paper. In addition, although Norway is a developed country, the country does not propose a net-zero emission target, but a carbon emission reduction target. Please refer to https://zerotracker.net/ for information on carbon neutral targets proposed by countries.

Point 4: Paris Agreement is mentioned in the “abstract” and after in the introduction but, then (lines 144 to 146) authors write: “(2) We compare and analyze the changes in each club’s emissions transition paths in the periods before and after the Kyoto Protocol came into force.” Although several of the Kyoto Protocol mechanisms are still in force, such as the carbon market, for example, the Kyoto Protocol failed to cut GHG emissions and perished with the Doha Amendment and all the other attempts until reaching the Paris Agreement. What are the reasons for not considering the Paris Agreement national commitments (NDC) and instead of insisting to explore the model with the old Kyoto Protocol?

Response : Thank you for your suggestion. In this study, we choose the entry into force of the Kyoto Protocol as the key event to observe the convergence change of per capita carbon emissions. On the one hand, it is because although most countries failed to achieve the targets set by the Kyoto Protocol within the stipulated time, the policies still played a certain effect of emission reduction after implementation. More importantly, the entry into force of the Kyoto Protocol meant that the world turned more attention to climate change, and countries began to take measures of different degrees to reduce carbon emissions, and laid the foundation for the implementation of the Paris Agreement at a later stage. On the other hand, because, the study period of this article is 1990-2019, while the Paris Agreement was signed in 2016, the time span of relevant carbon emission data is so short that it cannot support the club convergence analysis after the Paris Agreement.

Point 5: In lines 309 to 311 authors write “We posit that the convergence club division of PCCE in each country may change significantly in 2005, and the countries with reduced relative emissions levels can offer an advantageous policy reference for other countries.” What is the scientific base for such assumption knowing the history of Kyoto Protocol and its outcomes?

Response : Thank you for your suggestion. Although most countries have failed to meet the targets set by the Kyoto Protocol within the stipulated time, the policies have still had some effect in reducing emissions after implementation. More importantly, the entry into force of the Kyoto Protocol meant that the world turned more attention to climate change. Therefore, this article considers the entry into force of the Kyoto Protocol as a key event to observe the convergence change of per capita carbon emissions. After the Kyoto Protocol, countries started to take different degrees of measures to reduce carbon emissions, which made the convergence path of per capita carbon emissions in these countries change, and the countries whose relative emission levels reduced to a lower convergence club implied the adoption of carbon reduction policies, which can provide policy suggestions for other countries aiming at carbon neutrality. Finally, after the empirical results, it is shown that some of the countries required by the Kyoto Protocol to reduce their emissions did shift to the convergence club with lower relative emission levels. Please refer to section 3.2.

Point 6: On the other hand, in lines 301 to 304 authors state “Based on the club convergence of PCCE for each country in Figure 1, it is difficult to reflect the change in the carbon emissions levels among countries because of the long time span.” Please explain the difficulty.

Response : Thank you for your suggestion. With the relatively long time span of this study, the results of club convergence reflect the convergence process of per capita carbon emissions of each country within 30 years. The longer the time span of club convergence, the differences in carbon emissions per capita among countries are gradually ignored. Since the beginning of the new century, countries, especially developed Western countries, have made carbon emission reduction an important policy goal. With a longer time span, the club convergence method is difficult to capture the changes in the clubs that countries are in as a result of environmental policies (from clubs with lower emission levels to clubs with higher emission levels and from clubs with higher emission levels to clubs with lower emission levels). We have reworked the paragraph. Please see P313-P319.

Point 7: Concerning results: they are presented in a quite confusing manner expanding from an initial group of 4 ou 3 clusters presented in Table 1, passing through 11 convergence clubs (Table 2 – panel A) to a final set of 7 clubs. In all clubs, countries with quite different characteristics are positioned in the same group and this is not discussed in the paper. For example, Luxembourg is geographically quite a small country but a very high standard of living. Luxembourg is club 1 (table 1) alongside with Ethiopia, Iceland or Trinidad and Tobago. What relates these so different countries? In lines 273 to 275 authors write “Specifically, Club 1 primarily includes countries with large total carbon emissions, such as the US, high per capita GDP countries, such as the Netherlands, and energy producing countries, such as Saudi Arabia.” However, was the Netherlands part of the study? Club 1 of Table 1 does not include the Netherlands.

Response: Thank you for your suggestion. Due to error in data collation, the data of per capita carbon emissions for countries such as Ethiopia is wrong and the Netherlands is not included in our article, we are very sorry for this, we have finished the correction of the empirical results and we have made the changes, thank you very much for pointing out our mistake. Please see P270-P293 for the revised results and explanation of the club grouping.

Point 8: The analysis of the factors that were found to influence the clusters do not present any novelty: industrialization, energy intensity and GDP are factors that are well-known to positively influence carbon emissions.

Response : Thank you for your suggestion. Since this paper includes many developing countries with insufficient data, it is difficult to further adopt a more detailed and diverse perspective to analyze the factors affecting club convergence. Meanwhile, this article focuses on analyzing the differences in the changes of factors affecting club convergence before and after the formal entry into force of Kyoto Protocol, and the results show that with the formal entry into force of Kyoto Protocol, the effect of economic development level affecting PCCE gradually diminishes, and energy intensity gradually becomes an important factor affecting the convergence path of PCCE. Please refer to section 3.3.

Reviewer 4 Report

The paper “Club convergence of economies' per capita carbon emissions: Evidence from countries that proposed carbon neutrality” is an interesting manuscript that analyse the carbon emissions paths and steady-state 132 levels of countries that have proposed carbon neutrality targets, to construct an international club inventory of differing carbon emissions patterns and further analyse the driving factors affecting the formation of convergence clubs.   The paper is well written. The results are well presented and the discussion is exhaustive.   The paper could be accepted for publication on Int. J. Environ. Res. Public Health after minor revision. The main question is:

1) Format the text: in particular, authors should pay attention to when they write carbon dioxide because it must be written with the number 2 in subscript (CO2). This error occurs both in the text and in the references.

Author Response

Response to Reviewer 4 Comments

Point 1: The paper “Club convergence of economies' per capita carbon emissions: Evidence from countries that proposed carbon neutrality” is an interesting manuscript that analyse the carbon emissions paths and steady-state 132 levels of countries that have proposed carbon neutrality targets, to construct an international club inventory of differing carbon emissions patterns and further analyse the driving factors affecting the formation of convergence clubs. The paper is well written. The results are well presented and the discussion is exhaustive. The paper could be accepted for publication on Int. J. Environ. Res. Public Health after minor revision. The main question is: 1) Format the text: in particular, authors should pay attention to when they write carbon dioxide because it must be written with the number 2 in subscript (CO2). This error occurs both in the text and in the references.

Response : Thank you for your suggestion. We have thoroughly checked this article to eliminate formatting errors. Meanwhile, this article has been edited in English provided by MDPI.

Round 2

Reviewer 3 Report

2nd Review report of the manuscript IJERPH 1756364, with the title ‘Club convergence of economies' per capita carbon emissions: Evidence from countries that proposed carbon neutrality’.

The revised version of the manuscript IJERPH 1756364 received on July 1st, 2022 has corrected several errors and omissions that have been appointed in the 1st review report, including a thoroughly revision of English writing.

This article presents a club convergence model, based on the economic theory of convergence, applied to per capita carbon emissions (PCCE) of 121 countries in the period 1990 to 2019.

Based on the paper objectives (lines 101 – 105) which include the goal of “Understanding carbon emissions convergence patterns and the determinants of identified convergence clusters identified”, this manuscript suitability to the special issue "Decision Models for Sustainable Development in the Carbon Neutrality Era" appears to be just fair. Nevertheless, Sustainable Development is not an issue addressed in this manuscript.

Both the convergence model and its application to carbon emissions are not novel issues, however this paper presents historical perspective of countries’ PCCE trend until now. Authors adopted the Kyoto Protocol as the turning point for the analysis of the emissions trends of countries considering that this Protocol was the trigger for (lines 515 - 518) “…this article considers the entry into force of the Kyoto Protocol as a key event to observe the convergence changes of PCCEs. After the entry into force of the Kyoto Protocol, countries that signed the agreement may take different degrees of carbon reduction measures, which make the PCCEs convergence paths of these countries change,…”. The criterion chosen to include countries in the study was the existence of a country public pledge to achieve carbon neutrality in 2050 (lines 99 – 101).

Although several of the Kyoto Protocol mechanisms are still in force, such as the carbon market, for example, the Kyoto Protocol failed to cut GHG emissions and perished with the Doha Amendment and all the other attempts until reaching the Paris Agreement.

The argument presented by authors is quite feeble as the Kyoto Protocol finished 10 years ago and was only signed by 37 countries and the European Union meaning that less than 30% of the countries included in this study were not engaged by the Kyoto Protocol. China, the United States, India, or Brazil, some of the biggest polluters and GHGs emitters in the world, were never engaged with the Kyoto Protocol. I specifically used the word engaged instead of bound because, as it is well known, there were no consequences for those Parties that did not comply with their own objectives such was the case of Canada that abandoned the Kyoto Protocol just before its ending (December 2012) due to the total failure of Canada GHGs emissions reduction objectives. Therefore, besides the advantages of Kyoto Protocol mechanisms, its impact in the reduction of GHGs had, literally no effect other than in countries of the European Union.

At moment (2022), this study including 121 countries, using the Kyoto Protocol can only be considered as an historical summary and an exercise of mathematical adjustment of the convergence model. Through the application of this convergence model, countries with very different characteristics have been grouped in the same convergence club despite their huge diversity such as geographical dimension, geographical and climate zone, population density, natural resources potential, social policies, Human Development Index, poverty, lifestyles and standards and cost of living, among other countries’ aspects that are recognized to influence the PCCE.

In addition, there is a lack of a detailed discussion on “the determinants of identified convergence clusters identified”. Some of the explanations provided to such rare association of countries in convergence clubs are not supported by the knowledge of countries facts and reality. Therefore, outcomes of this manuscript are redundant as they bring no novelty or additional information to the current state of the art.

In the scope of the Open Science Policy guidelines, authors have the right to publish subjective perspectives of a theme if they clearly assume their hypotheses which is the case of this manuscript. As authors have corrected the omissions and errors found the 1st version of the manuscript, there are no objective reasons to oppose to the publication of this amended version of the manuscript, hoping that this work may constitute another motive to a valuable discussion about GHGs emissions reduction targets.